# HIV-1 Uncoating and Nuclear Import Precede the Completion of Reverse Transcription in Cell Lines and in Primary Macrophages

**DOI:** 10.3390/v12111234

**Published:** 2020-10-30

**Authors:** Ashwanth C. Francis, Mariana Marin, Mathew J. Prellberg, Kristina Palermino-Rowland, Gregory B. Melikyan

**Affiliations:** 1Department of Pediatrics, Division of Infectious Diseases Emory University School of Medicine, Atlanta, GA 30322, USA; mmarin@emory.edu (M.M.); m.prellberg@emory.edu (M.J.P.); kpalermino@gmail.com (K.P.-R.); 2Children’s Healthcare of Atlanta, Atlanta, GA 30322, USA

**Keywords:** virus imaging, HIV-1 capsid, uncoating, reverse transcription, nuclear import

## Abstract

An assembly of capsid proteins (CA) form the mature viral core enclosing the HIV-1 ribonucleoprotein complex. Discrepant findings have been reported regarding the cellular sites and the extent of core disassembly (uncoating) in infected cells. Here, we combined single-virus imaging and time-of-drug-addition assays to elucidate the kinetic relationship between uncoating, reverse transcription, and nuclear import of HIV-1 complexes in cell lines and monocyte-derived macrophages (MDMs). By using cyclophilin A-DsRed (CDR) as a marker for CA, we show that, in contrast to TZM-bl cells, early cytoplasmic uncoating (loss of CDR) is limited in MDMs and is correlated with the efficiency of reverse transcription. However, we find that reverse transcription is dispensable for HIV-1 nuclear import, which progressed through an uncoating step at the nuclear pore. Comparison of the kinetics of nuclear import and the virus escape from inhibitors targeting distinct steps of infection, as well as direct quantification of viral DNA synthesis, revealed that reverse transcription is completed after nuclear import of HIV-1 complexes. Collectively, these results suggest that reverse transcription is dispensable for the uncoating step at the nuclear pore and that vDNA synthesis is completed in the nucleus of unrelated target cells.

## 1. Introduction

The cone-shaped mature HIV-1 core encasing the viral ribonucleoprotein (vRNP) complex is an assembly of ~200–250 hexamers and 12 pentamers of the capsid protein (CA) [1,2]. Fusion between the virus and cellular membranes results in delivery of the viral core into the cytoplasm of target cells. Following cellular entry, the conical viral core is disassembled through a poorly understood process called uncoating, which is generally defined as a partial or complete loss of CA from the conical core (reviewed in [3]). Timely uncoating of the viral core is critical for the efficient reverse transcription of the vRNA genome into a viral cDNA (vDNA), which is mediated by the HIV-1 reverse transcriptase (RT) and takes place within the confines of the reverse transcription complex (RTC) [4,5]. Following the completion of reverse transcription, the long-terminal repeats (LTRs) of the vDNA ends are processed by the enzyme integrase (IN) to form the pre-integration complex (PIC), which is poised for integration [6,7]. Once inside the nucleus, IN catalyzes the strand-transfer step of vDNA integration into actively transcribing genes of host chromatin [8]. Here, we will use the term “viral replication complexes (VRCs)” to collectively represent the vRNP, RTC, and PIC forms of HIV-1 that cannot be distinguished by imaging assays [9].

The factors affecting HIV-1 uncoating include intrinsic core stability [10], reverse transcription [11,12,13,14], and CA–host-factor interactions that either stabilize or destabilize the capsid lattice (e.g., cyclophilin A, nucleoporins, cellular polyadenylation specific factor 6 (CPSF6), and TRIM5α) (reviewed in [3]). An intrinsic core stability is dictated by the strength of CA inter-subunit interactions and is modulated by amino acid substitutions in CA [10]. Interestingly, point mutations in CA that alter the core stability have been shown to elicit a defect in vDNA synthesis [10,15,16]. These results suggested a direct link between uncoating and reverse transcription. HIV-1 reverse transcription proceeds through a series of steps (reviewed in [17]) that appear to be completed over the course of several hours in different target cell lines [18,19,20,21,22]. Conversely, we and others have found that the initiation of vDNA synthesis promotes cytoplasmic uncoating [12,13,14]. This relationship has also been documented using a functional cyclosporine A (CsA) washout assay that monitors HIV-1 escape from the cytoplasmic restriction factor TRIMCyp that recognizes intact capsid lattice [11,20].

It has been thought that vDNA synthesis is completed in the cytoplasm, as evidenced by isolation of integration-competent PICs containing full-length HIV-1 DNA from the cytoplasm of infected cells [23,24,25,26,27]. Moreover, the synthesis of the central polypurine tract (cPPT), which occurs during the late stages of reverse transcription, has been proposed to modulate the nuclear import of HIV-1-based lentiviral genomes [28,29,30]. However, recent microscopy-based evidence suggested that reverse transcription is not required for HIV-1 nuclear import, since blocking vDNA synthesis did not impair the nuclear import of fluorescently tagged VRCs [19,31,32]. By contrast, the nuclear import of HIV-1 VRCs and the formation of 2-LTR circles, a surrogate marker for nuclear import of vDNA used in bulk assays [33,34,35], are efficiently blocked by small-molecule inhibitors targeting CA [36,37,38]. Moreover, CA and its interaction with host factors at the nuclear pore are the main determinants of HIV-1 nuclear import [39,40,41]. Therefore, irrespective of the effect of vDNA synthesis on early uncoating, a subset of functional CA assemblies retained by the VRCs interacts with the nuclear pore and facilitates HIV-1 nuclear import.

We have found that nuclear import of functionally relevant HIV-1 complexes progresses through a step of uncoating at the nuclear pore. We employed a cyclophilin A-DsRed (CDR) marker to label the viral core and detect a loss of CA from viral complexes labeled with IN-super-folder GFP (INsfGFP) in HeLa-derived TZM-bl cells [13,19]. Importantly, CDR binds CA with high avidity without drastically affecting virus infectivity or its interaction with host factors, and loss of CDR correlates with the loss of CA from cores during uncoating in vitro and in living cells [13,19]. Single-particle tracking in live cells revealed two distinct uncoating phenotypes: (1) early uncoating events, within the first hour or so after initiating infection, that were characterized by an abrupt and complete loss of CDR followed by proteasomal degradation of VRCs and (2) gradual loss of CDR from the remaining small fraction (<5%) of HIV-1 cores [13]. Our results showed that retention of nearly all CDR signal in the cytoplasm and a dramatic loss of this marker at the nuclear envelope was a prerequisite for HIV-1 nuclear entry leading to infection [19]. Collectively, these findings strongly argue against early and complete uncoating in the cytoplasm as a productive path to infection, in contrast to uncoating at the nuclear envelope. Importantly, uncoating at the nuclear envelope does not result in a complete loss of CDR or CA. A small number of CDR/CA molecules associated with HIV-1 complexes can be detected in the nucleus, and this signal disappears along with the VRCs upon integration of vDNA into host chromatin [19]. These results suggest that a terminal uncoating step occurs at the nuclear envelope. It is worth noting that most imaging studies [11,14,21,42,43], including our work [13,19], used HeLa-derived cell lines, while the early steps of HIV-1 infection in primary cells are not well characterized.

Human monocyte-derived macrophages (MDMs) are infected by HIV-1 in vivo and are relevant to viral pathogenesis (reviewed in [44]). The efficiency of HIV-1 infection in MDMs is largely determined by the need to mask the VRCs from innate immune sensing through CA–host-factor interactions [45,46,47] and by the low concentration of deoxynucleotide triphosphates (dNTPs) available for vDNA synthesis [48]. Low dNTP pools that result in inefficient reverse transcription of HIV-1 in these cells are maintained by the monocyte-specific restriction factor SAMHD1 [49]. Despite the inefficient vDNA synthesis in MDMs, single-virus imaging techniques have suggested that the conical core integrity is lost early during virus infection [14]. By contrast, other groups detected large amounts of immunolabeled CA signals associated with nuclear VRCs in MDMs [32,50]. Our recent study showed that the increased CA signals detected in nuclear VRCs do not originate from single cores, but from the clustering of multiple VRCs in nuclear speckle compartments of MDMs [9]. Moreover, a high concentration of the CA-targeting inhibitor PF74, which has been shown to modulate HIV-1 core stability and block vDNA synthesis in the cytoplasm, had negligible effects on preformed nuclear VRC-clusters in MDMs [9]. These observations suggest that nuclear VRCs are not enclosed in intact cores.

Here, using single-virus imaging in living cells, we show that the distinct types of HIV-1 uncoating (defined as loss of the CA marker CDR) previously documented in TZM-bl cells are also observed in terminally differentiated primary human MDMs. These include (a) abrupt loss of CDR early upon infection, which is promoted by vDNA synthesis, and (b) late uncoating at the nuclear envelope that can lead to productive nuclear import of VRCs [13]. However, compared to TZM-bl cells [13], a much larger fraction of post-fusion HIV-1 cores retain CDR for several hours in MDM cytosol, even after promoting vDNA synthesis by SAMHD1 depletion. Comparison of the kinetics of nuclear import of single functional VRCs and the kinetics of HIV-1 escape from inhibitors targeting CA and vDNA synthesis implied that reverse transcription continues past the CA-dependent steps of nuclear import in cell lines and MDMs. Collectively, our data suggest that although reverse transcription stimulates cytoplasmic uncoating, its completion is dispensable for the nuclear pore-associated uncoating step, which is a prerequisite for HIV-1 nuclear import.

## 2. Materials and Methods

### 2.1. Plasmids

The pHIV-eGFP, pR9ΔEnv, pCypA-DsRed (CDR), Vpr-IN-superfolder GFP (INsfGFP), and pMD2.G (VSV-G-expressing) plasmids have been described previously [13,19,51]. The psPAX2 vectors encoding lentiviral proteins were obtained from Addgene (Cat# 12260, a gift from Didier Trono). The pLenti.SNAP-LaminB construct has been described previously [9,19]. Plasmids encoding for SIV-Gag/GagPol Vpx(+) and SIV Vpx(−) were a kind gift from Dr. Nathaniel Landau [52].

### 2.2. Cell Lines and Reagents

The following reagents were obtained from the NIH AIDS Reference and Reagent Program, Division of AIDS, NIAID, NIH: pNL4-3.Luc.R-E- from Dr. Nathaniel Landau [53,54]; TZM-bl cells expressing CD4, CXCR4, and CCR5 from Drs. J.C. Kappes and X. Wu [55]; anti-p24 antibody AG3.0 donated by Dr. J. Alan [56]; RT inhibitor nevirapine and IN inhibitor raltegravir (Merck & Company, Kenilworth, NJ, USA).

HEK293T/17 cells (from ATCC, Manassas, VA, USA), CHME3 cells (a kind gift from Mojgan Naghavi [42]), and HeLa-derived TZM-bl cells were grown in high-glucose Dulbecco’s Modified Eagle Medium (DMEM, Mediatech, Manassas, VA, USA) supplemented with 10% fetal bovine serum (FBS, Sigma, St. Louis, MO, USA), and 100 U/mL penicillin–streptomycin (Gemini Bio-Products, Sacramento, CA, USA). HEK293T/17 cells were grown in a medium supplemented with 0.5 mg/mL G418 sulfate (Mediatech, Manassas, VA, USA). TZM-bl and CHME3 stable cell lines expressing EBFP2 or SNAP-LaminB1-10 were generated by transduction with a lentiviral vector pLenti.EBFP2-LaminB1-10 or pLenti.SNAP-LaminB1-10 encoding for LaminB1. Cells were subjected to limited dilution and selection of clones expressing optimal levels of fluorescent lamin.

Cyclosporin A (CsA) was from Calbiochem (Burlington, MA, USA). PF74 (#PF-3450074), aphidicolin (#A0781), Bright-Glo luciferase assay kit was from Promega (Madison, WI, USA). Antibodies against Cyclin T1 (mouse sc-271348) were from Santa Cruz Biotechnology, and mouse monoclonal primary antibody against CDK9 pS175 was a kind gift from Jonathan Karn (Case Western University). Cy5-conjugated anti-mouse antibody was from SouthernBiotech (Birmingham, AL, USA), and donkey anti-rabbit AF405 antibody (#ab175651) was purchased from Abcam (San Francisco, CA, USA). The SNAP-Cell 647-SiR dyes were from New England Biolabs (NEB, #S9102S). Phosphate buffered saline containing Mg^2+^/Ca^2+^ (dPBS) and Mg/Ca-free (PBS) was from Corning (MediaTech, Manassas, VA, USA). EdU (5-ethynyl-2′-deoxyuridine) (#A10044), Click-iT EdU Imaging kits (#C10338 and #C10340) and Hoechst 33342 (#62249) was from Thermo Scientific.

### 2.3. Pseudovirus Production and Characterization

Fluorescently labeled pseudoviruses were produced and characterized as described previously [13]. Briefly, HEK293T/17 cells grown in 6-well culture plates were transfected with the following plasmids: VSV-G (0.2 μg), pHIVeGFP (2 μg), Vpr-INsfGFP (0.5 μg), and/or CypA-DsRed (0.5 μg). SIV VLPs encoding or not encoding Vpx (Vpx(+) or Vpx(−)) were generated by transfecting respective Gag-Pol plasmids [52] (2 μg) and VSV-G (0.2 μg). For generating SNAP-Lamin-expressing lentivirus, pLenti.SNAP-LaminB1-10 (2 μg) was cotransfected with the HIV-1 Gag-Pol-expressing psPAX2 (1 μg) and VSV-G (0.2 μg) vectors.

Six hours after transfection, the medium was replaced with 2 mL of fresh DMEM/10% FBS without phenol red, and the samples were incubated for an additional 36 h at 37 °C, 5% CO_2_. Viral supernatant was collected, filtered through a 0.45 μm filter, and quantified for p24 content using AlphaLISA immunoassay kit (PerkinElmer, Waltham, MA, USA) or RT activity (RTU) measured using the PERT protocol as described previously [19,57]. For live-cell imaging, fluorescent viruses were purified through a 20% sucrose cushion or concentrated 10× using LentiX concentrator (Clontech Laboratories, Inc., Mountain View, CA, USA). Concentrated viruses were resuspended in FluoroBrite (GIBCO) or RPMI-1640 medium containing 10% FBS, aliquoted, and stored at −80 °C. MOI was determined in TZM-bl cells by examining the percentage of eGFP-expressing cells after 48 h of infection with VSV-G-pseudotyped HIVeGFP virus.

### 2.4. Isolation, Differentiation and Treatment of MDMs

Human peripheral blood mononuclear cells (PBMCs) were isolated from de-identified volunteer donor blood samples for the preparation of MDMs. PBMCs were isolated from fresh heparinized blood by Ficoll–Hypaque gradient centrifugation and monocytes were isolated by magnetic labeling using Monocyte Isolation Kit II (Miltenyi Biotec Inc, Bergisch Gladbach, Germany) according to the manufacturer’s protocol. Enriched monocytes were differentiated to MDMs in RPMI-1640 supplemented with 10% FBS, 100 µg/mL streptomycin, 100 U/mL penicillin, 2 mM glutamine, and 5 ng/mL GM-CSF (Gemini Bio-Products 300124P020G70K) and maintained in cytokine-supplemented medium for 7 days, as described previously [9]. Imaging experiments were performed between 1 and 21 days after GM-CSF removal. Where indicated, SAMHD1 was depleted by treating monocytes (immediately after collection) with SIV VLPs Vpx(+) or Vpx(−) containing 10 RT units (RTU) followed by differentiation in RPMI-1640 medium containing GM-CSF. The SNAP-LaminB nuclear envelope marker was expressed by transduction of Vpx(+)-treated monocytes [9]. SAMHD1 depletion was verified at 1, 7, and 14 days post GM-CSF removal by fluorescence microscopy following immunostaining with primary anti-SAMHD1 mouse polyclonal antibody (Abcam, cat# ab67820) and secondary Cy5-conjugated anti-mouse antibody from SouthernBiotech (Birmingham, AL, USA). Nearly all cells treated with Vpx(+) showed efficient SAMHD1 depletion as described in [9].

### 2.5. Single-Cycle Infection Assay

A single round infectivity assay was performed in 96-well black-plates (Corning, Kennebunk, ME, USA). Ten thousand cells/well of terminally differentiated MDMs or CHME3 or TZM-bl cells were plated 24 h prior to infection with VSV-G-pseudotyped HIV-1 at MOI 0.5 (MDMs) or MOI 0.2 (TZM-bl and CHME3 cells). Following spinoculation with virus, cells were cultured at 37 °C for 120 h (MDMs) or 48 h (TZM-bl and CHME3) and lysed, and luciferase activity was measured using the Bright-Glo luciferase substrate (Promega). Where indicated, the drugs raltegravir (10 µM), nevirapine (10 µM), or PF74 (2.5 µM) were added at indicated times after infection. Image-based analysis of infected cells was performed in an 8-well chamber glass slide using a VSV-G-pseudotyped HIVeGFP or pR9Δ virus. The percentage of infected cells was obtained by normalizing the number of infected cells expressing either the eGFP reporter or immunostained for cytoplasmic Gag to the total number of Hoechst-labeled nuclei from (n > 1000 from 4 random fields of view for each independent experiment).

### 2.6. Fixed-Cell Imaging and Immunofluorescence Assay

Fixed-cell imaging experiments in MDMs were performed at MOI 0.5–5 (determined in TZM-bl cells), as indicated in figure legends. Fluorescent viruses were bound to cells by spinoculation at 16 °C, as above, and washed once in dPBS. Where indicated, EdU (5 μM) or HIV-1 inhibitors were added to cells prior to shifting to 37 °C to initiate virus entry, followed by cell fixation at different times post-infection. MDMs were fixed with 4% PFA (Electron Microscopy Sciences, #1570-S) for 7 min followed by vDNA detection using the manufacturer’s protocol. Immunostaining was performed as described previously [9] using primary antibody AG3.0 against CA/p24 (diluted 1:100), anti-LaminB1 antibody (1:1000), anti-CycT1 (1:200), anti-CDK9-pS175 (1:200), and secondary goat anti-mouse Cy5 antibodies (1:1000). After incubation with primary antibodies, cells were washed five times and incubated with goat anti-rabbit AlexaFluor405 or AF555 antibodies (1:1000). Primary antibodies were added in a blocking solution (0.5% BSA) overnight at 4 °C, and secondary antibodies were incubated in blocking buffer for 1 h at room temperature. Five washing steps in blocking solution were used to remove nonspecific binding.

### 2.7. Live-Cell Imaging of HIV-1 Uncoating, Nuclear Entry, and Infection

HIV-1 uncoating and nuclear import in live cells were visualized as previously described [19]. In brief, VSV-G-pseudotyped HIV-1 particles colabeled with INsfGFP and CypA-DsRed were bound to 5·10^5^ differentiated MDMs. In these cells, the fusion and early uncoating steps were imaged between 0 and 2 h, the late uncoating was imaged between 2 and 16 h, and the nuclear import steps were imaged between 0 and 40 hpi. Cells were infected with MOI 0.008 and 0.5 for early uncoating and late uncoating or nuclear import experiments in MDMs, respectively. MOI 0.2 was used for imaging uncoating, nuclear import, and infection of aphidicolin (10 μM)-treated TZM-bl-SNAP-Lamin or CHME3-EBFP2-Lamin cells between 0 and 24 hpi. Where mentioned, MDM nuclei were stained for 10 min with 2 μg/mL Hoechst 33342. Alternatively, TZM-bl-SNAP-Lamin cells or Vpx(+)-treated MDMs expressing the SNAP-Lamin nuclear envelope marker were labeled for 30 min with SNAP-Cell 647-SiR dyes (NEB, #S9102S) prior to virus binding. Following spinoculation, the cells were washed twice, and virus entry was initiated by adding prewarmed complete RPMI-1640 medium (Gibco) to MDMs or complete FluoroBrite imaging medium (Gibco) to TZM-bl and CHME3 cells in an imaging chamber mounted on a temperature- and CO_2_-controlled microscope stage.

### 2.8. Image Acquisition

Live cell imaging was performed on a Zeiss LSM780 or LSM880 laser scanning confocal microscope, using a C-Apo 40×/1.2NA water-immersion or 63×/1.4NA oil-immersion objective. To visualize early fusion or uncoating events occurring within 2 hpi, 11–15 Z-stacks spaced by 1 μm were acquired every 20–30 s. Twenty-five neighboring fields of view were imaged by tile scanning to increase throughput. Long-term imaging of live cells was performed by acquiring 11–15 Z-stacks spaced by 1 or 0.7 μm for every 5–7 min (slow acquisition, when imaging for 40 h) or every 40–90 s (fast acquisition, when imaging for 12–18 h), as indicated. A DefiniteFocus module (Carl Zeiss) was utilized to correct for axial drift. Nuclei stained with Hoechst or SNAP-Lamin and containing INsfGFP and CypA-DsRed were imaged using highly attenuated 405, 488, 561, and 633 nm laser lines. When imaging fixed cells, greater laser powers and line averaging were used, along with more stringent axial sampling (~45 Z-stacks spaced by 0.3 μm) to improve axial resolution and signal-to-background ratio. 3D-image series were processed off-line using ICY image analysis software (http://icy.bioimageanalysis.org/) [58].

### 2.9. Single-Particle Tracking and Image Analysis

The initial annotation of HIV-1 uncoating and INsfGFP complex entry into the nucleus was done by visually examining time-lapse movies. After inspection, software-assisted single-particle tracking was used to analyze viral complexes in the cytoplasm and determine uncoating, the time of arrival at the nuclear envelope, and the time of penetration/import into the nucleus. Time-lapse 3D images were converted to maximum intensity projections and single-particle tracking was performed in 2D using the ICY image analysis platform. Single-particle tracking was performed using INsfGFP as a reference signal and the background-subtracted fluorescence intensities of CDR and SNAP-Lamin were plotted after normalization to respective initial fluorescence intensities. 2D INsfGFP objects were detected and tracked using the spot detection and spot-tracking plugins. The docking, uncoating, and nuclear entry times were determined manually for those events that were not amenable to single-particle tracking.

An in-house protocol was created using the ICY protocols module to discriminate between cytoplasmic and nuclear INsfGFP spots [9]. After spatial segregation of IN-labeled HIV-1 complexes and background subtraction, intensity analysis was performed, as described previously [9]. Colocalization was determined as the fraction of intranuclear INsfGFP spots containing above-background levels of CypA-DsRed, EdU/vDNA, Cyclin T1, or CDK9/pS175 signals.

### 2.10. Statistical Analyses

Statistical significance was determined using a nonparametric Mann–Whitney rank-sum test or Student’s *t*-test. Unless otherwise indicated, *p* < 0.05 (^∗^) was considered significant; ^∗∗^ and ^∗∗∗^ denote *p* < 0.01 and *p* < 0.001, respectively. The number of experiments and error bars are indicated in the figure legends.

## 3. Results

### 3.1. Productive HIV-1 Uncoating and Nuclear Import Precede the Completion of Reverse Transcription in Cell Lines

We sought to visualize key steps of single HIV-1 infection—docking at the nuclear envelope, uncoating, and nuclear import followed by integration and reporter gene expression—and correlate these events with the acquisition of resistance to inhibitors targeting distinct steps of infection, using a bulk infectivity assay. Toward this goal, we colabeled VSV-G-pseudotyped HIVeGFP reporter viruses with INsfGFP and CDR, as previously described [19,59]. This labeling strategy has allowed us to image single HIV-1 transport to the nuclear envelope (NE), uncoating (loss of CDR), nuclear import, and disappearance of the INsfGFP puncta (a correlate of integration)—all in the context of HIV-1 reporter gene expression [19]. Imaging these early steps of HIV-1 infection in HeLa-derived TZM-bl cells revealed that (1) single HIV-1 uncoating leading to infection occurs at the NE and (2) loss of nuclear INsfGFP signal, which was observed for 25% of nuclear IN puncta, correlates well with subsequent expression of the eGFP reporter. Importantly, the disappearance of nuclear INsfGFP is effectively blocked by inhibitors of HIV-1 integration [19,43,60]. In contrast, cells in which nuclear INsfGFP puncta remained visible throughout the prolonged imaging experiment did not get infected.

Cells were infected at a low MOI (0.2), and imaging or time-of-drug-addition experiments were performed for a duration of 22 h. Here, we focused on productive nuclear import of single INsfGFP complexes that disappeared in the nucleus culminating in infection. As shown in Figure 1A,B, single-virus infection progressed through a loss of the CDR at the nuclear envelope, followed by intranuclear transport of VRCs and subsequent disappearance of a fraction of these VRCs, which correlated with subsequent expression of the eGFP reporter. At the same time, the lack of disappearance of nonproductive nuclear HIV-1 VRCs allowed us to distinguish productive integration events and back-track the nuclear import step of IN-labeled VRCs that disappeared in the nucleus and established infection (Figure 1A, also see Appendix A).

All 111 tracked INsfGFP particles that disappeared in the nucleus (out of 430 total nuclear import events) exhibited a loss of CDR after VRC docking at the NE (Figure 1B). In contrast, INsfGFP/CDR-labeled HIV-1 cores that docked at the nuclear envelope for several hours without a significant loss of CDR failed to enter the nucleus (Appendix A). These cores were eventually displaced into the cytoplasm, exhibiting long-range anterograde trafficking towards the cellular periphery (Appendix A; see also Appendix A), further supporting the notion that uncoating at the NE is a prerequisite for VRC import into the nucleus [19]. Importantly, the ability to visualize the uncoating, nuclear import, and subsequent disappearance of single VRCs (a surrogate for integration [19,43,60]) that establish infection enabled us to measure, for the first time, the kinetics of these important steps in the context of productive entry of HIV-1 (Figure 1C).

We next compared the kinetics of nuclear import and subsequent INsfGFP disappearance with the kinetics of HIV-1 escape from inhibitors targeting distinct stages of virus infection measured using a time-of-drug-addition assay. The CA-targeting inhibitor PF74 (2 μM), the non-nucleoside reverse transcription inhibitor nevirapine (NVP, 10 μM), and the integrase strand transfer inhibitor raltegravir (RAL, 10 μM) were added at different times post-infection, and the resulting infection was measured after 48 h (Figure 1C). HIV-1 escape from NVP and RAL addition reflects the kinetics of completion of vDNA synthesis and the integration steps, respectively. By contrast, at a comparatively low concentration (2 μM), the CA–hexamer binding inhibitor PF74 [61,62,63] alters the core stability [9,13,64] and inhibits VRC nuclear import [9,19,22,32,50,61,62,63], without affecting vDNA synthesis [9,20,22,50]. In addition, by competing with the CPSF6 binding to CA assemblies in the nucleus, PF74 also interferes with VRC transport to nuclear speckles [9]. Thus, the time-course of virus escape from PF74 reflects the completion of the above-mentioned CA-dependent steps of infection.

Whereas the kinetics of INsfGFP disappearance and escape from RAL were virtually superimposable, as expected [19], the completion of reverse transcription was much slower compared to the nuclear import kinetics or the virus escape from PF74 (Figure 1C). The half-time of nuclear import of the integration-competent complexes was ~4 h, which is shorter than the half-time of escape from PF74 (~6 h), completion of vDNA synthesis (~8 h), or the half-time of integration (~12 h) (Figure 1C). While the delayed kinetics of HIV-1 escape from PF74 relative to the nuclear import kinetics is in agreement with the role of CA after nuclear entry (transport of VRCs to nuclear speckles), the faster kinetics of PF74 compared to NVP addition further supports the notion that CA-dependent nuclear transport steps are also completed prior to the completion of vDNA synthesis.

A similar progression through distinct steps of infection was observed in human microglial CHME3 cells. In these cells, CDR loss at the NE was followed by nuclear trafficking and INsfGFP puncta disappearance in the nucleus leading to infection (n = 53, Appendix A). In these cells, the half-time of nuclear import (~4 h) was shorter than the half-time of escape from PF74 (~6 h) and NVP (~8 h). The half-time of IN disappearance was similar to that of RAL addition (~10 h), as expected (Appendix A). Collectively, these observations suggest that the nuclear import of VRCs that go on to establish infection progresses through a loss of CDR at the nuclear envelope and this step precedes the completion of vDNA synthesis in different target cell lines (Figure 1C and Appendix A).

### 3.2. Reverse Transcription Continues Past HIV-1 Escape from CA-Targeting Inhibitor and Is Completed in the Nucleus of MDMs, Independent of SAMHD1 Depletion

Next, we examined HIV-1 entry and uncoating in terminally differentiated human MDMs, which are relevant targets for in vivo infection. We first verified that CDR incorporation into HIV-1 did not affect its infectivity in MDMs. Pseudoviruses labeled with CDR were as infectious as unlabeled viruses (Appendix A). The addition of CsA, which binds CypA and displaces it from CA, reduced the infectivity in MDMs by more than 5-fold (Appendix A). These results are consistent with the notion that CA interaction with CypA is important for HIV-1 infection of MDMs [45]. As reported by other groups [49,52], depletion of SAMHD1 by treatment of MDMs with Vpx-containing SIV VLPs (abbreviated Vpx(+)) resulted in a marked (up to 10-fold) increase of HIV-1 infection (Appendix A), apparently due to increased cytoplasmic dNTP levels. The infectivity-enhancing effect of SAMHD1 depletion was independent of whether the viruses were labeled with CDR. These results demonstrate that CDR, which tightly binds to the CA lattice through high-avidity interactions [13,19], is a noninvasive marker to study single HIV-1 uncoating in cell lines and primary human monocyte-derived cells [9].

To gain functional insights into the progression of HIV-1 infection in MDMs, we performed time-of-drug-addition experiments over an extended time period of up to 5 days. The single-cycle infectivity was measured by a luciferase assay and normalized to the DMSO control. In both untreated and Vpx(+)-treated MDMs, the overall kinetics of completion of vDNA synthesis appeared rate-limiting, as it was closely followed by HIV-1 integration (Figure 2A,B). In the presence of SAMHD1 (untreated cells), vDNA synthesis continued even after 4 days post-infection (Figure 2A). The lack of plateau for the kinetics of viral escape from NVP and RAL (Figure 2A) shows that, in untreated MDMs, reverse transcription and integration continue past 100 hpi. By contrast, in Vpx(+)-treated cells, in which HIV-1 infection was improved by >10-fold over control (Appendix A), both the kinetics of completion of vDNA synthesis and integration were accelerated, reaching plateau by about 70 hpi (Figure 2B). Interestingly, the kinetics of HIV-1 escape from PF74 were comparable for untreated and Vpx-transduced MDMs, reaching a plateau at about 48 hpi (Figure 2). The estimated half-time of HIV-1 escape from 2 μM PF74 (~12 h) was much shorter than the respective half-times of completion of vDNA synthesis in untreated (>48 h) and Vpx(+)-treated (~24 h) cells (Figure 2A,B). The faster kinetics of HIV-1 escape from PF74 compared to the escape from NVP addition suggests that CA-dependent steps (uncoating, nuclear import, and association with nuclear speckles) precede the completion of reverse transcription [9], independent of the level of SAMHD1 expression in MDMs.

It is noteworthy that, although the kinetics of completion of vDNA synthesis was accelerated in Vpx(+) MDMs, the observed half-time (~24 h) of reverse transcription was still much longer compared to a half-time of ~8 h observed in cell lines (Figure 1C and Appendix A). These results are in line with the observation that the intracellular dNTP pool in SAMHD1-depleted cells is ~5-fold lower than the dNTP levels detected in cell lines [65,66,67].

### 3.3. SAMHD1 Depletion in MDMs Diminishes the Pool of Stable Post-Fusion Cores

To determine if reverse transcription stimulates uncoating, we infected MDMs with a low MOI of labeled pseudoviruses that yielded 40–60 INsfGFP/CDR-colabeled cores per cell. Single-particle tracking performed between 0 and 2 h post-infection revealed that, similar to our observations in TZM-bl cells [13], a fraction of INsfGFP-labeled cores quickly lost CDR (early uncoating) in MDMs (Appendix A). The total number of post-fusion cytoplasmic cores was determined by adding CsA at 80 min post-infection. This treatment selectively displaces CDR from cytoplasmic cores (Figure 3A,B and Appendix A) but does not release CDR from intact viruses residing on the cell surface or in endosomes [13,59,68]. Analysis of CDR loss revealed that only a minor fraction (<20%) of cytoplasmic cores underwent uncoating and these events were completed by 40 min post-infection (Appendix A), while the remaining 80% of stable post-fusion cores retained the CDR marker until CsA addition (Appendix A). In contrast to inefficient uncoating in MDMs, VSV-G-mediated pseudovirus fusion was quite efficient and was nearly completed within the first hour of infection (Appendix A), as determined in parallel experiments by the addition of CsA during virus entry. These results suggested that the overwhelming majority of cytoplasmic cores remain stable in MDMs, likely due to the limited reverse transcription in these cells (Figure 2A).

By contrast, depletion of SAMHD1 by Vpx(+) treatment of MDMs promoted early uncoating, as evidenced by a significant increase in the fraction of INsfGFP complexes losing CDR (Appendix A) and a decreased in the fraction of stable post-fusion cores detected by CsA addition at 80 min (Figure 3A,B and Appendix A; see also Appendix A), as compared to control cells treated with Vpx(−) SIV-VLPs or untreated cells (Appendix A). These results support correlation between the HIV-1 core stability and the efficiency of reverse transcription in MDMs. It is worth noting that, even in Vpx(+)-treated MDMs, the fraction of post-fusion cores (~40%) retaining CDR beyond 80 min (hereafter referred to as “long-lived cores”) is much greater than in TZM-bl cells (<5%) [13].

### 3.4. Long-Lived HIV-1 Cores Retain CDR over the Course of Several Hours Irrespective of SAMHD1 Depletion

We have previously shown that early uncoating in TZM-bl cells leads to VRC degradation in proteasomes, meaning that these events are unlikely to culminate in infection [13,19]. Accordingly, irrespective of whether HIV-1 uncoating occurred early or late in MDMs, the loss of CDR in the cytoplasm was usually followed by disappearance of INsfGFP complexes (Appendix A). We therefore quantified the fraction of INsfGFP/CDR-colabeled puncta over time as a readout for the core stability. Consistent with the efficient CDR incorporation into HIV-1 (>85% of INsfGFP-labeled viruses incorporated CDR [9,19]) and degradation of post-uncoating IN complexes [13,19], the fraction of INsfGFP puncta lacking CDR at all time points was negligible (less than 5%). The measurement of the number of IN/CDR-colabeled puncta over time revealed that, even in Vpx(+) MDMs, a significant fraction of long-lived HIV-1 cores retained robust CDR signal at 16 hpi (Figure 3C; also see Appendix A). Ensemble averaging of single-particle fluorescence of isolated long-lived cores that were trackable over 10 h revealed little or no apparent loss of CDR signal in the MDM cytoplasm (beyond the photobleaching controls) in either untreated or Vpx(+)-treated cells (Figure 3D and Appendix A).

Together, these results demonstrate a much greater HIV-1 core stability in MDM cytoplasm compared to TZM-bl cells. Although this marked difference may reflect a somewhat lower dNTP concentration in Vpx(+) MDMs compared to cell lines [65,66,67], we favor the interpretation implicating other cell type-specific host factors in modulating the HIV-1 core stability in the cytoplasm.

### 3.5. HIV-1 Nuclear Import in MDMs Progresses through a Loss of CDR at the Nuclear Envelope and Is Accelerated upon SAMHD1 Depletion

In order to visualize nuclear import of VRCs in MDMs, time-lapse imaging of untreated and Vpx(+)-treated cells infected with INsfGFP/CDR-labeled viruses was performed every 5 min up to 40 hpi. HIV-1 nuclear import was measured based on the appearance of INsfGFP puncta in the nucleus, as previously described [19]. Under live cell imaging conditions, the CDR signal associated with nuclear INsfGFP puncta was low or undetectable, supporting the notion that uncoating preceded the nuclear import step in MDMs, similar to TZM-bl cells [19]. Moreover, we were unable to visualize the integration and eGFP expression in MDMs using low MOI (0.5 measured in TZM-bl cells) amenable to single-particle tracking, even upon SAMHD1 depletion. This observation is consistent with our finding that HIV-1 infection progresses over several days in MDMs, and < 30% of SAMHD1-depleted cells become infected by 5 days post-infection with identical virus input [9]. We therefore analyzed the uncoating step by back-tracking the nuclear INsfGFP puncta to their respective sites of docking at the NE.

Single-particle tracking in MDMs is challenging due to the high degree of mobility exhibited by the majority of these cells (see also Appendix A). As is the case for migratory cells [69,70], MDM nuclei frequently rotated and changed shape, thus greatly confounding reliable single-particle tracking with respect to the nuclear envelope. For this reason, we were able to faithfully track only a subset (<30%) of HIV-1 nuclear import events in MDMs, unlike the readily trackable nuclear import in less mobile TZM-bl cells (Figure 1, see also [19]). The tracked nuclear import events in MDMs were mainly observed in less mobile cells and for complexes that docked at the nuclear envelope for longer than 5 min (the temporal resolution of our time-lapse imaging). For all tracked nuclear import events in MDMs, INsfGFP puncta lost CDR signal while colocalized with the NE, prior to entering the nucleoplasm (Figure 4A,B; see also Appendix A and [9]).

Live-cell image analysis of HIV-1 nuclear import revealed that the import kinetics in untreated MDMs was much slower than in Vpx(+)-treated cells. On average, 50% of HIV-1 IN-labeled VRCs entered the nucleus of untreated cells within ~11.5 hpi (n = 57), whereas in Vpx(+)-treated MDMs the half-time of nuclear import was ~5.8 h (n = 157) (Figure 4C). However, neither the fraction of MDMs containing at least one nuclear INsfGFP complex (~30% at MOI of 0.5) nor the number of nuclear INsfGFP puncta observed in these cells showed a significant difference between untreated and Vpx(+)-treated MDMs (Figure 4D,E). In MDMs, in which HIV-1 nuclear import was detected, on average, only ~1.5 INsfGFP puncta were observed per nucleus under our experimental conditions (Figure 4E). In comparison, a similar virus input resulted in 4–8 INsfGFP complexes entering the nucleus of TZM-bl cells with a half-time of ~4 h [19]. These results show that, although Vpx(+) treatment accelerates the kinetics of nuclear import, this treatment is without effect on the extent of HIV-1 nuclear entry, likely because fewer cores survive in the cytoplasm of SAMHD1-depleted cells (Figure 3A–C). Importantly, regardless of the presence of SAMHD1, HIV-1 nuclear import was largely completed prior to the completion of vDNA synthesis (compare Figure 2 and Figure 4C), suggesting that the final steps of reverse transcription take place inside the nucleus.

### 3.6. Improved HIV-1 Infection in SAMHD1-Depleted MDMs Correlates with a More Efficient Nuclear Reverse Transcription

To obtain further evidence for reverse transcription in the nucleus, we infected MDMs with INsfGFP-labeled pseudoviruses at a higher MOI of 2. This virus input resulted in infection of ~20% of untreated and 60–80% of Vpx(+)-treated MDMs, respectively [9], and also allowed delivery of a sufficient number of VRCs into the nucleus for the detection of associated vDNA. The reverse-transcribed vDNA in VRCs was visualized by in situ labeling with the nucleoside analog 5′-ethynyl deoxyuridine (EdU), as reported by others and us [9,22,32,50,71]. We have recently shown that, under these experimental conditions, multiple nuclear HIV-1 VRCs traffic towards the nuclear speckle compartments in MDMs where they merge to form stable multi-VRC clusters [9]. We thus analyzed the increase in INsfGFP and EdU/vDNA signals of nuclear VRCs over time as a measure of their clustering and correlated this to productive infection. Infection was defined by the appearance of diffuse cytoplasmic staining for Gag after 36 hpi in Vpx(+)-treated MDMs (Figure 5A). We also note that nuclear clusters persisted after Gag-expression, consistent with their reported stability in MDMs [9].

Measurements of INsfGFP puncta fluorescence over time revealed a similar increase in signal in Vpx-transduced and untreated MDMs, irrespective of NVP treatment (Figure 5B). This result implies that HIV-1 nuclear import and clustering occur independently of reverse transcription. In spite of a drastic increase in the nuclear INsfGFP signal over time, the average number of nuclear IN puncta remained constant (~4, Figure 5C), which is consistent with our finding that HIV-1 complexes accumulate in approximately four nuclear speckle compartments found in MDMs [9].

Notably, clustering of nuclear INsfGFP puncta reached a near completion by ~36 hpi, as indicated by the plateau in the kinetic curve (Figure 5B), whereas the EdU signal that was associated with these IN complexes continuously increased over time (Figure 5D). This signal, as expected, was significantly increased by Vpx treatment and abrogated by NVP treatment (Figure 5D). Thus, consistent with the recent findings [9,72], vDNA synthesis can continue beyond HIV-1 nuclear import and VRC clustering. The kinetics of infection (onset of Gag-expression) closely followed that of vDNA synthesis and reached ~20% and ~80% by day 4 in untreated and Vpx(+) MDMs, respectively (Figure 5A,D,E). These results collectively suggest that the robust infection of SAMHD1-depleted MDMs is likely due to a more efficient vDNA synthesis in the nucleus.

### 3.7. Host Transcriptional Factors Are Enriched at the Sites of vDNA-Containing Nuclear HIV-1 Clusters

The persistence of INsfGFP-labeled VRC clusters after Gag expression and their association with EdU/vDNA in the nucleus of infected MDMs (Figure 5) prompted the investigation of HIV-1 transcription at these sites. Towards this goal, we analyzed the recruitment to nuclear HIV-1 clusters of Cyclin T1 and Cyclin-Dependent Kinase 9 (CDK9) phosphorylated at Serine 175 (CDK9-pS175), which are the components of the positive transcription elongation factor b (P-TEFb) complex (reviewed in [73]). Immunostaining of MDMs for components of the P-TEFb complex recruited to transcribing viral genomes revealed enrichment of both Cyclin T1 and CDK9-pS175 at nuclear VRC clusters (Figure 6). Interestingly, both DMSO- and RAL-treated samples showed similar enrichment of Cyclin T1 and CDK9-pS175 P-TEFb signals in multiple donors, in agreement with a recent report [71], and this enrichment was blocked by NVP treatment (Figure 6C–E). The fraction of EdU/vDNA-positive VRC clusters colocalizing with the P-TEFb complex was 63–78% across donors and showed minor differences in staining pattern (Figure 6F). Under these conditions, ~54% of DMSO-treated and 5% of RAL-treated MDMs became infected, as evidenced by Gag/GagPol expression (Figure 6G). These results are in good agreement with our previous observation that a significantly lower number of vRNA puncta was associated with nuclear VRC clusters in the presence of RAL [9]. Taken together, these results suggest that while P-TEFb may associate with nuclear HIV-1 clusters, vDNA integration into the host genome is required for efficient viral transcription. Inefficient transcription may occur from unintegrated vDNA templates in RAL-treated cells [74,75]. These data are in good agreement with the finding that vDNA integration into nuclear speckle-associated genomic domains occurs at the location of nuclear HIV-1 clusters [9].

## 4. Discussion

Here, we provide evidence that vDNA synthesis is, in large part, completed in the nucleus of target cells. We showed this by (1) directly visualizing continued nuclear vDNA synthesis in MDMs (see also [9]), which progressed more efficiently in the absence of SAMHD1 (Figure 5), and (2) corroborating this result in TZM-bl and CHME3 cells by correlating the kinetics of single HIV-1 nuclear import and productive integration with the completion of vDNA synthesis and infection assessed by a time of drug addition assay (Figure 1C and Appendix A). In agreement with this notion, late vDNA transcripts are found predominantly in the nucleus of target cells [18,76]. Our current work thus confirms and expands upon the recent reports that reverse transcription is completed in the nucleus [9,18,21,72,77]. Another critical finding of this study is the observation of terminal HIV-1 uncoating (loss of CDR) at the nuclear envelope in cell lines and primary MDMs.

Reverse transcription appears to stimulate uncoating [11,12,13,14]. However, the cellular sites and extent of uncoating that allow for the completion of vDNA synthesis remain unknown. We have previously found that not all post-fusion viral cores in a given cell uncoat in the same manner [13]. Cores either abruptly (early uncoating) or gradually (late uncoating) lost CDR in the cytoplasm of TZM-bl cells [13]. Here, we report similar HIV-1 uncoating phenotypes in primary MDMs (Figure 3B and Appendix A). We find that HIV-1 nuclear import progresses through a late uncoating step at the NE in MDMs and cell lines. However, a much larger fraction of cores were stable in the cytoplasm of MDMs compared with TZM-bl cells, even after depletion of SAMHD1 in MDMs which facilitates vDNA synthesis and infection by raising the intracellular dNTP levels. These results imply that although the uncoating phenotypes appear to be preserved across different cell types, the determinants of the stability of infection-establishing long-lived HIV-1 cores are cell type-dependent.

Interestingly, in addition to destabilizing the HIV-1 cores, SAMHD1 depletion accelerated the kinetics of nuclear import in MDMs (Appendix A) but had no effect on the extent of VRC nuclear import (Figure 4D,E). It is currently unclear whether SAMHD1 has a direct role in delaying HIV-1 nuclear import or if the accelerated nuclear import in SAMHD1-depleted cells is an indirect effect of Vpx(+) treatment. Regardless of the presence of SAMHD1 in MDMs, vDNA synthesis was completed after nuclear import and was greatly delayed compared to that in TZM-bl and CHME3 cells (Figure 1, Figure 2 and Figure 4, and Appendix A, also see [19]). These results are consistent with the higher dNTP levels in cell lines compared to those in SAMHD1-depleted MDMs [65,66,67]. Importantly, we observed a more efficient reverse transcription in the nuclear VRCs of SAMHD1-depleted MDMs (Figure 5D). Collectively, these results suggest that while reverse transcription promotes cytoplasmic uncoating, the improvement of HIV-1 infection in SAMHD1-depeleted MDMs is likely due to more efficient vDNA synthesis and completion in the nucleus.

Single-virus tracking experiments in the context of infection have suggested that the HIV-1 core integrity is lost early during infection [14]. Consistent with this observation, core-associated eGFP-Vpr appears to dissociate from the cytoplasmic cores shortly after fusion [78] and is not detected in the nuclear VRCs [79]. Additional evidence for an early loss of core integrity comes from the observation that the chimeric cytoplasmic Lens Epithelial Derived Growth Factor (LEDGF/p75) IN-binding domain-eGFP (IBD-eGFP) fusion protein expressed in target cells effectively binds IN of incoming vRNPs and blocks HIV-1 integration [80,81]. We have found that VRCs that abruptly lost all CDR in the cytoplasm lack CA/p24 signal and are targeted for proteasomal degradation [19]. By contrast, cores docked at the NE contained all or a large fraction of the CA marker CDR which was lost prior to HIV-1 nuclear import in MDMs and cell lines. In TZM-bl and CHME3 cells, all single HIV-1 nuclear import and integration events were preceded by loss of CDR at the NE [19]. Taken together, these findings suggest that uncoating of long-lived cores that culminates in a marked loss of CDR/CA at the NE likely underlies productive HIV-1 infection. It is worth stressing that direct or indirect imaging of CA molecules associated with the HIV-1 core lacks the sensitivity to detect minor losses of CA in the cytoplasm prior to docking at the nuclear pore where the terminal loss of core-associated CA takes place. Accordingly, the loss of some CA in the cytoplasm appears to be required for HIV-1 interaction with microtubule adaptor proteins [82] and host factors at the nuclear pore [83]. The mechanism for partial core disassembly and retention of fragments of CA lattice remains poorly defined. Recent studies demonstrated that inositol hexaphosphate (IP6) that is packaged into virions [84,85,86] can stabilize CA hexamers and help retain a partial CA shell during gradual uncoating.

Regardless of the extent of HIV-1 uncoating in the cytoplasm or at the NE, a subset of CA molecules, as well as CDR, remains associated with VRCs after import into the nucleus [9,19,20,21,32,43,50,59,87,88,89,90]. These CA molecules play a critical role in recruiting the host-factor CPSF6 to transport VRCs to sites of integration in actively transcribing genes [9,19,21,32,40,41,87] that are enriched in nuclear speckles [9]. Whether a partial CA lattice is sufficient for the intranuclear HIV-1 transport to nuclear speckles is not known. While fluorescence imaging lacks the resolution to study the conformational changes in the conical core, techniques, such as correlative fluorescence and electron microscopy (CLEM), have been successfully applied to gain structural insights into HIV-1 uncoating [88,91]. Collectively, these studies suggested that while intact or nearly intact cores can be accommodated within the nuclear pore complex, their passage into the nucleoplasm results in a dramatic restructuring of the viral core [88,91] and loss of the vRNP-associated electron density [91]. These studies provide proof of concept that partial CA assemblies might be retained after the loss of the integrity of CA lattice; however, the relevance of these CA-containing empty structures to infection remains unclear.

A recent study using an alternative CA marker, eGFP-CA, concluded that HIV-1 uncoats in the nucleus [21]. HIV-1 cores containing a small amount of eGFP-CA (1:15 ratio of eGFP-CA to unlabeled CA) did not exhibit detectable changes in fluorescence upon nuclear import, but eGFP-CA was lost in the nucleus, near the sites of virus integration. This loss of eGFP-CA signal in the nucleus, which was augmented by PF74 treatment, was interpreted as uncoating of intact or nearly intact cores. However, in contrast to this report, we did not observe a destabilizing effect of PF74 on nuclear CA/p24 signal [9]. In addition, a small fraction of eGFP-CA (~7% of unlabeled CA) incorporated into the viral core [21] may not faithfully report the fate of the unlabeled CA molecules forming the capsid lattice. Indeed, another study using a similar CA-GFP marker concluded that this probe is associated with the vRNP residing inside the capsid shell [43]. If the latter is true, one would not expect to see a loss of such vRNP-associated probe upon nuclear import, but this probe will likely be lost at the time of HIV-1 integration. Accordingly, we have consistently observed co-disappearance of the nuclear VRCs colabeled with INsfGFP and CDR during integration [19]. This result suggests that a terminal uncoating step occurs at the nuclear pore complex, whereas the intranuclear disappearance of the CDR remnants is a consequence of VRC disassembly during integration. Improvement of CA labeling approaches will be needed to delineate the extent of CA loss in the cytoplasm, at the NE, and inside the nucleus.

## Figures and Tables

**Figure 1 viruses-12-01234-f001:**
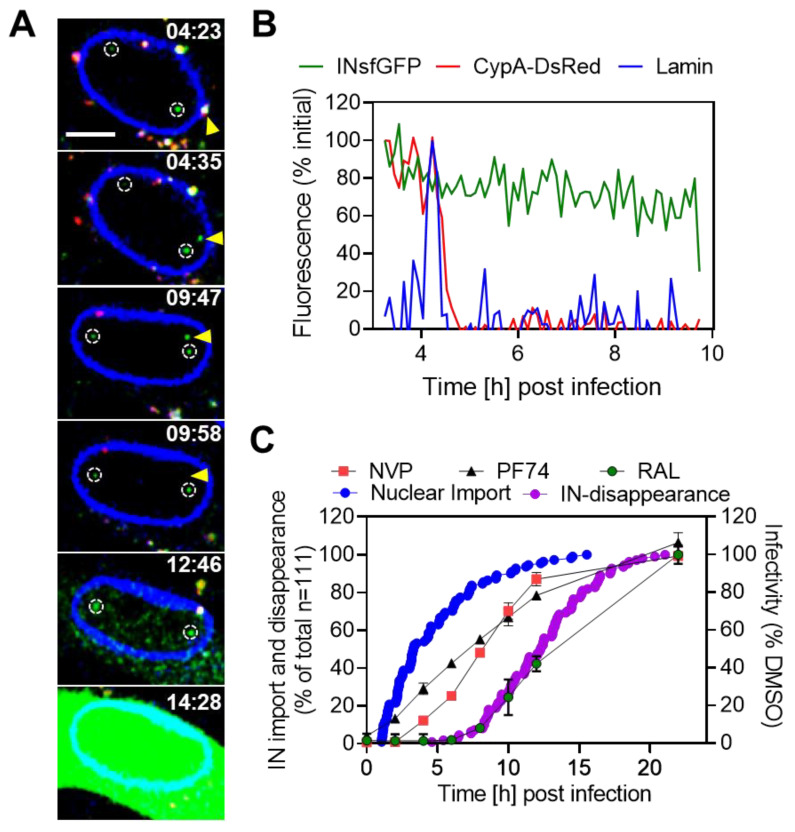
Time course of HIV-1 uncoating, nuclear import, reverse transcription, and integration in TZM-bl cells. HIV-1 uncoating, nuclear import, and integration in the context of infection in TZM-bl cells were visualized by live-cell imaging. (**A**) Images and (**B**) fluorescence intensity traces of single HIV-1 integrase (IN) complex (green) marked by arrow in (**A**) entering the nucleus of a TZM-bl cell after losing cyclophilin A-DsRed (CDR)/capsid assembly (CA) signal (red) at the nuclear envelope labeled by EBFP2-Lamin (blue). Inside the nucleus, the IN-superfolder GFP (INsfGFP) complex traffics for several hours prior to disappearing (a correlate of integration, marked by an arrowhead) at 12 h 20 min, which is followed by expression of HIV-encoded eGFP reporter starting at 14 h 15 min. The two additional nuclear INsfGFP puncta (marked by dashed lines) that do not disappear in the nucleus likely represent nonproductive nuclear entry events. Scale bar in (**A**) is 5 µm. (**C**) Comparison of the time courses of HIV-1 nuclear import, completion of viral cDNA (vDNA) synthesis, and integration. The kinetics of nuclear import (blue circles) and INsfGFP disappearance (green circles) in TZM-bl cells were obtained by live-cell imaging, as shown in (**A**,**B**). The kinetics of completion of reverse transcription and integration were determined in parallel experiments by adding nevirapine (NVP, 10 µM), PF74 (2 µM), and raltegravir (RAL, 10 µM) at indicated time points after infection. The resulting infection (luciferase signal) is plotted on the right axis. Data in (**C**) are represented as cumulative of n = 111 nuclear import and disappearance events obtained from live-cell experiments and means and SEM for drug addition experiments are from 3 independent experiments.

**Figure 2 viruses-12-01234-f002:**
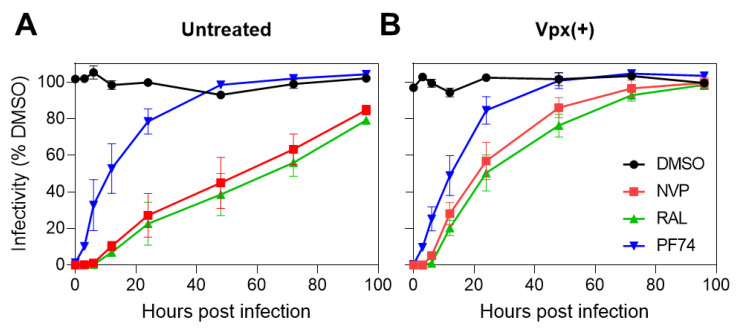
Time course of early steps of HIV-1 infection in monocyte-derived macrophages (MDMs) assessed by time of inhibitor addition. Untreated (**A**) and SIV-VLP Vpx(+)-treated MDMs (**B**) were infected with VSV-G-pseudotyped pNL4.3R-E-luc HIV-1 at an MOI of 2. Inhibitors of reverse transcription (nevirapine (NVP) 10 µM), capsid–host-factor interactions (PF74 2.5 µM), and integration (raltegravir (RAL) 20 µM) were added at indicated time points after synchronized infection (see Section 2). Results are normalized to DMSO control. Single round infectivity was measured at 5 days post-infection by luciferase assay. Error bars represent STD from triplicate experiments from 3 donors.

**Figure 3 viruses-12-01234-f003:**
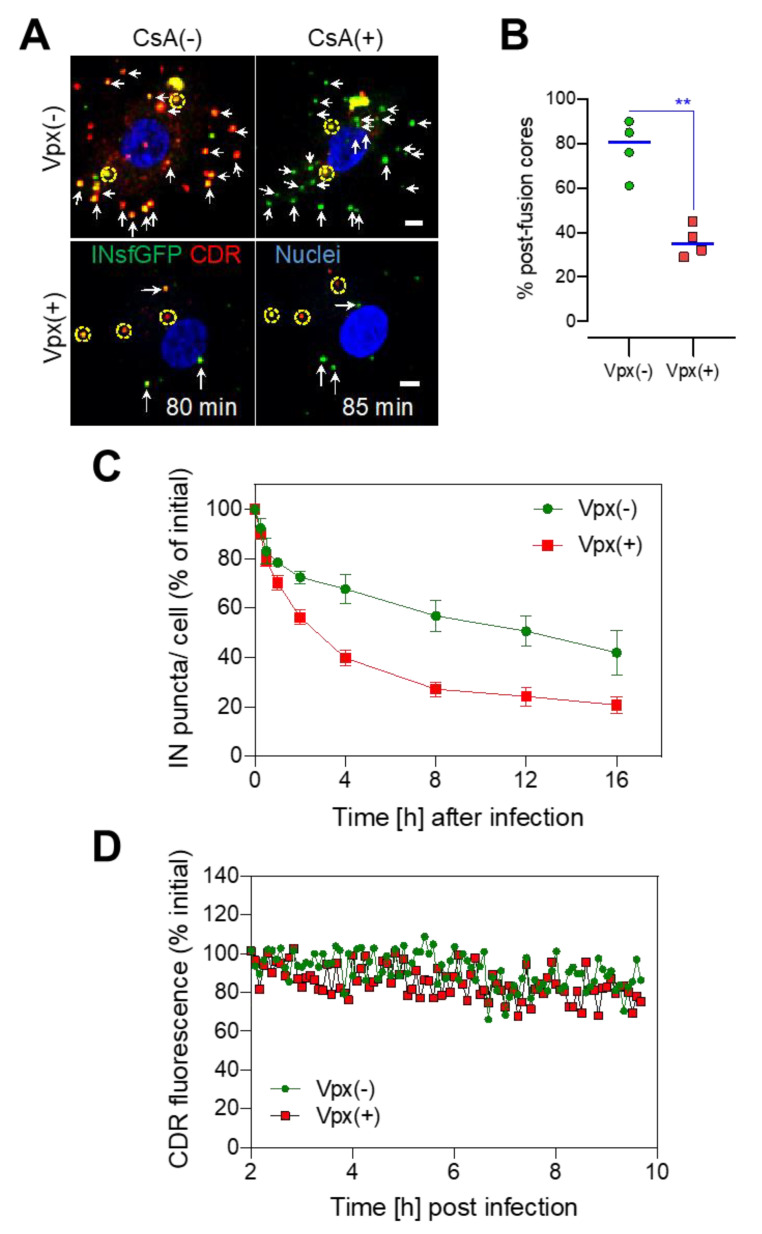
SAMHD1 depletion accelerates early cytoplasmic uncoating. MDMs treated with Vpx(+) or Vpx(−) SIV-VLPs were infected at MOI 0.5 with VSV-G-pseudotyped HIV-1 colabeled with INsfGFP (green) and CDR (red). (**A**) Shown are images of Vpx(−)- and Vpx(+)-treated MDMs at 80 min post-infection taken immediately before (CsA(−), left panel) and 5 min after cyclosporine A (CsA) addition (CsA(+), right panel) which resulted in a loss of CDR from post-fusion cytoplasmic viral cores (arrows) and not from intact virions (dashed yellow circles). Scale bar 5 μm. (**B**) The fraction of post-fusion cores surviving beyond 80 min post-infection in MDMs treated with Vpx(−) and Vpx(+) SIV-VLPs was determined by subtracting the number of viruses that did not lose CDR after CsA addition and normalizing to the initial number of cell-bound viruses at T = 0 min. Mean values (blue lines) from multiple experiments for 4 donors (data points) are shown. (**C**) The fraction of INsfGFP-labeled HIV-1 complexes that retained CDR signal in the MDM cytoplasm was normalized to the initial number of cell-associated viruses. Error bars are SEM from 20 randomly selected cells. (**D**) Average CDR fluorescence intensity of cores in the MDM cytoplasm imaged for 8 h starting at 2 hpi. Data show that regardless of Vpx(+) treatment HIV-1 cores may retain the CDR for a long time. Tracks are ensemble averages of n = 3 and n = 4 randomly selected cytoplasmic cores in Vpx(−)- and Vpx(+)-treated MDMs, respectively. Error-bars are STD. ^∗∗^ denote *p* < 0.01, respectively.

**Figure 4 viruses-12-01234-f004:**
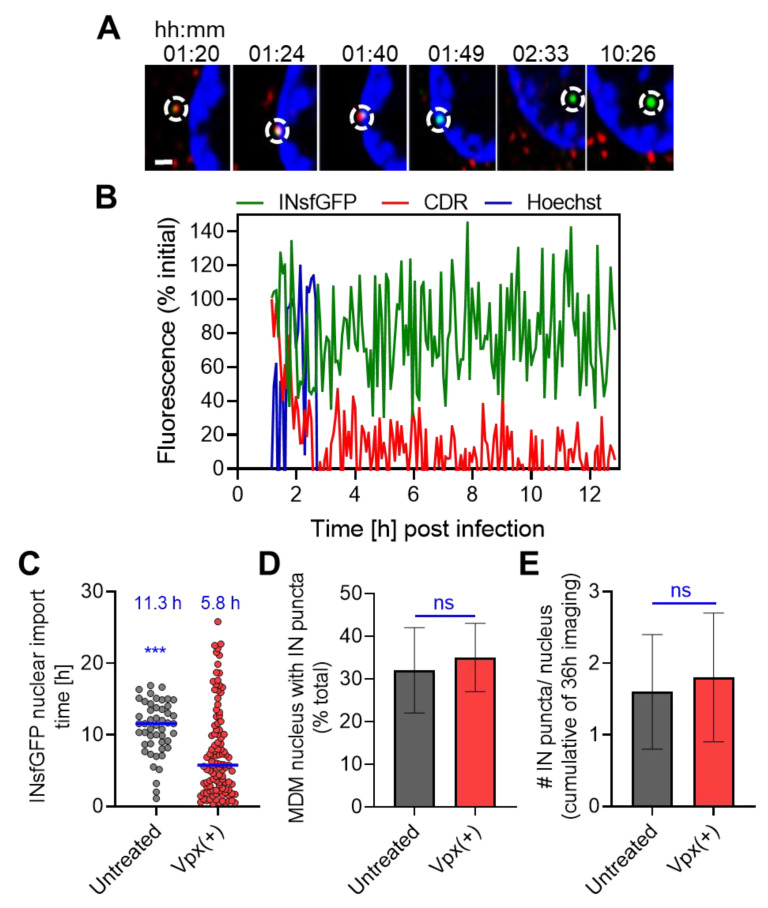
HIV-1 nuclear import is preceded by the loss of CDR/CA marker at the nuclear envelope and is accelerated by Vpx(+) treatment. Untreated or SIV-VLP Vpx(+)-treated MDMs were infected at MOI 0.5 with VSV-G-pseudotyped HIVeGFP virus fluorescently labeled with INsfGFP and CDR. Single-virus tracking was performed to visualize HIV-1 uncoating and nuclear import from 0 to 40 hpi. (**A**,**B**) An example of single HIV-1 nuclear import in MDM progressing through a loss of CDR at the nuclear membrane and intranuclear transport of the INsfGFP complex. (**A**) Images and (**B**) single-particle fluorescence intensity traces of HIV-1 uncoating and nuclear import in Vpx(+)-treated MDM showing a major loss of CDR signal (red) from INsfGFP viral replication complexes (VRCs) (green) at the nuclear envelope (blue) and retaining above-background CDR signals in the nucleus. Scale bar in (**A**) is 2 μm. (**C**) The kinetics of HIV-1 nuclear import in MDMs untreated or treated with Vpx(+). The kinetics of HIV-1 nuclear entry was obtained by manually annotating the time of appearance of IN puncta inside the nucleus observed in multiple experiments by time-lapse imaging in the ranges of 0–2, 0–6, 2–12, or 0–40 hpi, using 30, 60, 120, or 300 s intervals between image acquisitions, respectively. N = 57 and N = 195 nuclear entry of IN puncta were visualized for untreated and Vpx(+), respectively, from >5 donors. The half-times of nuclear import are shown in blue. (**D**,**E**) Effect of Vpx transduction of MDMs on HIV-1 nuclear import within 40 h of infection. (**D**) The fraction of untreated and SIV-VLP Vpx(+)-treated MDMs that contained at least one nuclear INsfGFP complex. (**E**) The average number of INsfGFP puncta per nucleus. N > 700 and N > 4000 nuclei from 5 and 12 independent experiments in untreated and Vpx(+)-treated MDMs, respectively. ^∗∗∗^ denote *p* < 0.001, respectively.

**Figure 5 viruses-12-01234-f005:**
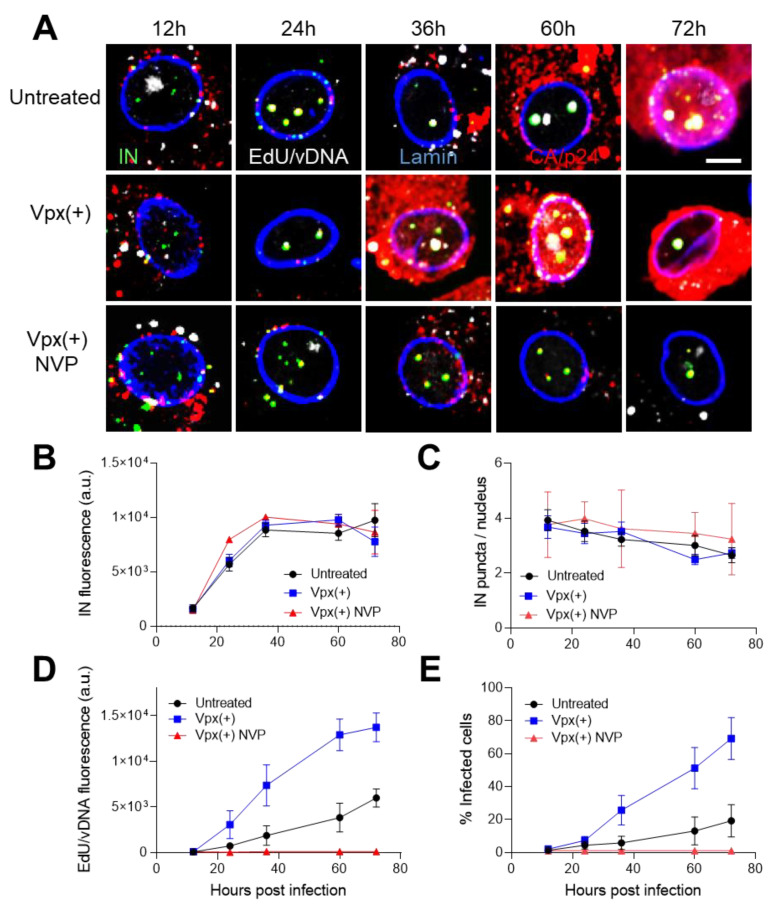
HIV-1 cDNA synthesis in MDMs continues past nuclear import and clustering of VRCs. MDMs pretreated with SIV-VLP Vpx(+) or left untreated were infected with VSV-G-pseudotyped HIV-1 labeled with INsfGFP (MOI 5) in the presence of EdU (5 μM). Cells were cultured for 72 h without removing the virus from the medium, fixed at the indicated time points, and stained for EdU and CA. Where indicated, the reverse transcriptase (RT) inhibitor nevirapine (NVP, 10 μM) was added to Vpx(+)-treated MDMs at the time of infection. (**A**) Representative images of MDM nuclei at different times post-infection showing INsfGFP (green), EdU/vDNA (white), CA/p24 (red), and lamin (blue). Diffuse CA signal in the cytoplasm appearing at 36, 60, and 72 hpi in Vpx(+) and at 60 and 72 hpi in untreated samples is a result of Gag/GagPol synthesis in productively infected cells. MDM nuclei were analyzed for INmNG fluorescence (**B**), the number of IN puncta per nucleus (**C**), IN puncta-associated EdU signal (**D**), and the fraction of infected cells (**E**), as a function of time. Scale bar in (**A**) is 5 μm. Error bars in (**B**–**E**) are SEM and mean from 30–40 nuclei for each condition and are representative for 2 donors.

**Figure 6 viruses-12-01234-f006:**
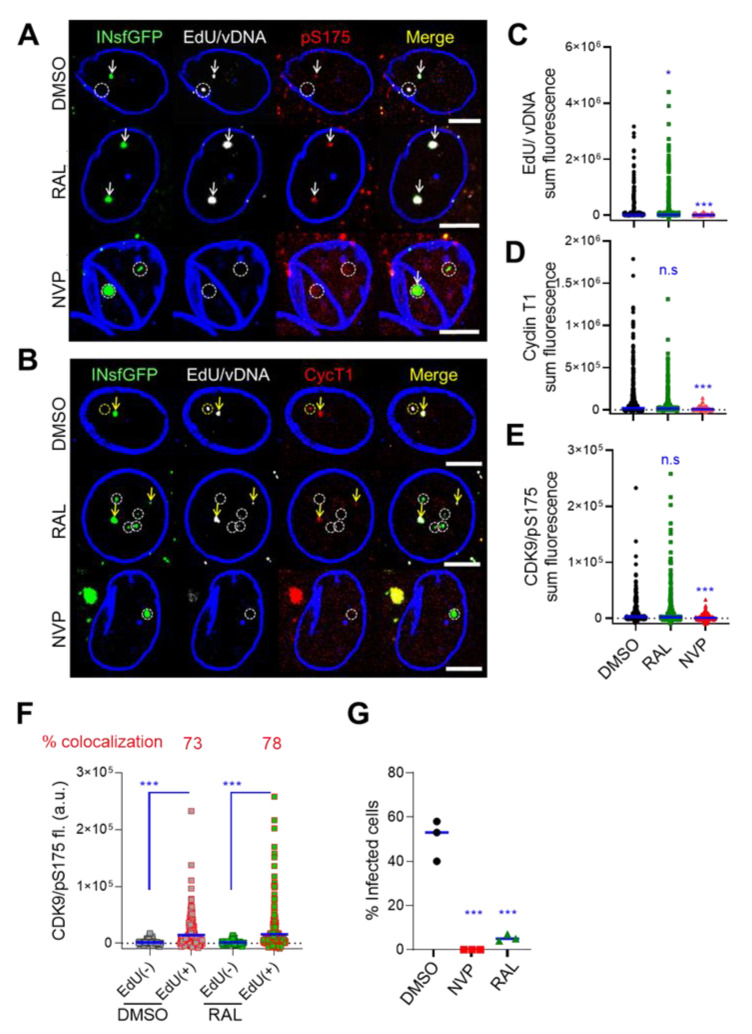
Cellular transcription machinery associates with nuclear HIV-1 clusters. MDMs were pretreated with Vpx(+) SIV-VLPs and infected with VSV-G-pseudotyped HIV-1 labeled with INsfGFP by spinoculation (MOI of 2). Unbound virus was washed away, EdU (5 μM) was added, and MDMs were cultured for 72 h. (**A**–**E**) As indicated, infections were carried out in the presence of nevirapine (NVP, 10 μM) or raltegravir (RAL, 20 μM). Cells were fixed and stained for EdU/vDNA and the components of the P-TEFb transcription complex, Cyclin T1 (CycT1) and CDK9/pS175 (pS175). (**A**,**B**) Images of MDM nuclei showing INsfGFP fluorescence (green), EdU/vDNA (white), and pS175 (**A**) or CycT1 (**B**) staining (red). Arrows in (**A**) and (**B**) point to the IN spots colocalized with EdU and P-TEFb complex; dashed circles indicate IN clusters that do not colocalize with EdU, P-TEFb, or both. (**C**–**E**) Analysis of nuclear INsfGFP puncta colocalization with EdU (**C**), Cyclin T1 (**D**), and CDK9/pS175 (**E**). (**F**) Analysis of CDK9/pS175 signal in INsfGFP-labeled VRCs that colocalize with (EdU(+)) or do not colocalize (EdU(−)) with EdU signal. (**G**) The fraction of eGFP-expressing infected cells in control (DMSO) or NVP (10 μM)- or RAL (10 μM)-treated samples was determined for 3 donors. Scale bar in (**A**) and (**B**) is 5 μm. Blue lines in (**C**–**G**) represent SEM and median at 95% CI representative of > 300 nuclear IN puncta from >50 nuclei (**C**–**F**) and percentage of infected cells from 3 donors (**G**). *p* < 0.05 (^∗^) was considered significant; ^∗∗∗^ denote *p* < 0.001, respectively.

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
