# Peer review of "HIV-1 Uncoating and Nuclear Import Precede the Completion of Reverse Transcription in Cell Lines and in Primary Macrophages"

_viruses, 2020, doi:10.3390/v12111234_

Round 1
Reviewer 1 Report
The submitted manuscript entitled “HIV-1 uncoating and nuclear import precede the completion of reverse transcription in cell lines and in primary macrophages” conserns a topic of high interest in HIV-1 field. This is particularly important for this new era of RT. Actually the RT has been discovered 50 years ago and only in 2020 some studies clarify the spatiotemporal action of this process.
Said that there are some points that should be improved before acceptance of the manuscript.
- The ref. 8 is line 40 is not appropriate since it concerns viral nuclear entry and not transcription.
- There is a problem with the number of references, in fact most of the references are wrong. I think there are lost references but the numbers are left alone. Please correct it as it is essential to the quality of the manuscript which is difficult to assess at the current state.
Abstract revisions:
- Line 24: The authors means the end of reverse transcription because the process already starts in the cytoplasm.
Introduction revisions:
- The sentence (line 81-82) is not accurate. In fact, there are new studies, such as Blanco-Rodriguez et al., JVI, 2020 that shows the remodelling of the viral core during the nuclear translocation using primary CD4+ T cells. In the paper this reference has been incorrectly included in line 84.
- The word uncoating should be better explained. In the last sentence do the authors refer to a complete uncoating phase? If so, this is not appropriate because the tools used are not sensitive enough to support this conclusion. I would ask you to correct this concept in different parts of the manuscript when necessary.
Results revisions:
- The viral system used by the authors is based on viral particles carrying in trans the INsfGFP. Thus, the system used is based on the image of the transgenic protein that competes with the WT IN. Is the INsfGFP functional? How the authors can be sure that they are not following just the fluorescent protein? Is the virus detected by INsfGFP is functional for the integration?
- Authors should make it clear in the text that the imaging system used is a live imaging approach that can provide important information, however this approach is not sensitive enough unlike other types of microscopes which have better resolution. This implies that the authors can easily miss the formation of new viral forms of CA derived from a partial uncoating at the pore. The authors should clarify this point.
- In the experiments described in figure 1 and in figure 2 there is a condition missed. High dose of PF74. In fact, it has been described by the group of Yamashita that PF74 has a bimodal behavior dose dependent. The addition of this condition in both experiments will certainly strenghten the results.
- Line 530 the results indicating that HIV-1 nuclear import and clustering occur independently of reverse transcription confirmed results published by others that have not been mentioned, please introduce the ref here (Rensen et al., BioRxiv 2020).
- The authors should better interpret and discuss the results showed in figure 6F. The % of co-localization of the vDNA with CDK9/pS175 is similar with or without RAL. However, episomal forms have a really low level of transcription. So how the authors can combine these results with their hypothesis that the detected EdU nuclear spots represent sites of active viral transcription?
Discussion revision:
- Is it correct to write “in large part” in line 580? I could not find any quantitative data in the paper.
- Line 588 two references are missed (Selyutina et al., Cell Reports, 2020; Rensen et al., BioRxiv, 2020).
- Line 642: Correctly the authors discussed that rare viral nuclear events caracterized by intact cores have been detected inside the nucleus using sophisticated microscopy tools. Correctly the authors mentioned that these rare structures can be empty structure. However, the authors miss to discuss another recent paper (Blanco et al., JVI, 2020) that shows open CA structure chains containing the retrotranscribed genome translocating through the NPC. These viral structures have been visualized using CLEM. In parallel the authors should discuss that several groups were able to visualize the viral CA associated with the viral DNA inside the nucleus (Peng et al., 2014). The biochemical presence of the viral CA protein was shown for the first time in the nucleus of macrophages and HeLa cells by Zhou and colleagues. The presence of CA in the nucleus was shown also by IF by several other groups (BejaranoDA, et al., 2019; Zurnic Bonisch I et al., 2020).
- The discussion should be better organized as there are several repetitions that could be avoided to help the reader.
Author Response
Reviewer #1:
The submitted manuscript entitled “HIV-1 uncoating and nuclear import precede the completion of reverse transcription in cell lines and in primary macrophages” conserns a topic of high interest in HIV-1 field. This is particularly important for this new era of RT. Actually the RT has been discovered 50 years ago and only in 2020 some studies clarify the spatiotemporal action of this process.
Said that there are some points that should be improved before acceptance of the manuscript.
- The ref. 8 is line 40 is not appropriate since it concerns viral nuclear entry and not transcription.
We thank the reviewer for catching this mistake. The reference 8 in line 41 has been changed. We now cite the original paper from the Bushman group [1].
- There is a problem with the number of references, in fact most of the references are wrong. I think there are lost references but the numbers are left alone. Please correct it as it is essential to the quality of the manuscript which is difficult to assess at the current state.
We apologize for this formatting error. To support the statements regarding host protein involvement in uncoating, the references15-37 (lines 46-47) have been removed and replaced with a review by Campbell and Hope (2015) [2] that discusses these articles,.
Abstract revisions:
Line 24: The authors means the end of reverse transcription because the process already starts in the cytoplasm.
Our data presented in Fig. 5 and results reported by others in primary MDMs and in cell lines [3-5] suggest that reverse transcription itself is dispensable for uncoating and nuclear import, but that, normally, reverse transcription is completed in the nucleus.
Introduction revisions:
- The sentence (lines 81-82) is not accurate. In fact, there are new studies, such as Blanco-Rodriguez et al., JVI, 2020 that shows the remodelling of the viral core during the nuclear translocation using primary CD4+ T cells. In the paper this reference has been incorrectly included in line 84.
To our knowledge, a few papers have looked at CA/p24 localization in CD4+ T-cells and primary MDMs, whereas the majority of studies investigated HIV-1 uncoating in cell lines. Thus, the statement in Introduction that “most imaging studies … used HeLa-derived cells …” (lines 84-87) appears to be correct. We thank the reviewer for catching the mistake with improper referencing of Ref 64 [6]. This is now moved to discussion along with other CLEM papers (lines 628-634).
- The word uncoating should be better explained. In the last sentence do the authors refer to a complete uncoating phase? If so, this is not appropriate because the tools used are not sensitive enough to support this conclusion. I would ask you to correct this concept in different parts of the manuscript when necessary.
We define uncoating as follows (lines 33-35): “Following cellular entry, the conical viral core is disassembled through a poorly understood process called uncoating, which is defined as a partial or complete loss of CA from the conical core (reviewed in [2])”.
We explain the interpretation of uncoating from our data in lines 70-85 by revising lines 80-85 as follows: “These findings strongly argue against early and complete uncoating in the cytoplasm as a productive path to infection, in contrast to uncoating at the nuclear envelope. Importantly, uncoating at the nuclear envelope does not result in a complete loss of CDR or CA. A small number of CDR/CA molecules associated with nuclear HIV-1 complexes disappears along with the VRCs upon integration of vDNA into host chromatin [7]. These results suggest that a terminal uncoating step occurs at the nuclear envelope”.
Accordingly, lines 112-114 refer to the NPC-associated uncoating step that is not influenced by reverse transcription: “Collectively, our data suggest that, whereas reverse transcription stimulates cytoplasmic uncoating, its completion is dispensable for the nuclear pore-associated uncoating step, which is a pre-requisite for HIV-1 nuclear import.”
Given that uncoating may progress through several steps involving a gradual loss of CA molecules [7-9], the timing of its completion is contentious. While fluorescence imaging in live cells cannot reveal structural changes in the conical core, changes in the signal from a CA marker, CypA-DsRed, can be reliably monitored by live-cell imaging techniques. On the contrary, EM techniques can only show snapshots of structural changes, but cannot provide time-resolved information regarding the nature and functional relevance of partial CA assemblies in the nucleus and address whether these complexes undergo further uncoating. Our experiments suggest that uncoating, as measured by loss of CypA-DsRed, is complete at the NPC and the remaining CypA-DsRed-labeled CA detected in the nucleus is lost along with the VRC upon integration [7]. We discuss this in lines: 605-652.
Results revisions:
- The viral system used by the authors is based on viral particles carrying in trans the INsfGFP. Thus, the system used is based on the image of the transgenic protein that competes with the WT IN. Is the INsfGFP functional? How the authors can be sure that they are not following just the fluorescent protein? Is the virus detected by INsfGFP is functional for the integration?
The incorporation of a fluorescent IN protein using Vpr- or Gag-based fusion proteins into Gag/GagPol encoding virions has been extensively validated and shown to represent a functional viral replication complex competent for integration by our group [7,9-11] and by others [3,5,8,12-18]. Moreover, we have shown that INsfGFP may participate in integration reaction, as they can functionally complement an inactive D64N/D116N catalytically inactive IN mutant derived from full-length Gag/GagPol [10], however less efficiently than when combined with WT IN from Gag/GagPol. Thus, INsfGFP which forms a stable complex with WT IN-containing PICs and is a faithful marker for HIV-1 nuclear import. Furthermore, the disappearance of INsfGFP in the nucleus of a living cell reports an integration event, as evidenced by abrogation of the IN disappearance events in infected cells treated with Raltegravir [7,12,18] and the overlapping kinetics of IN disappearance and virus escape from Raltegravir added at varied times (Fig. 1C) .
2. Authors should make it clear in the text that the imaging system used is a live imaging approach that can provide important information, however this approach is not sensitive enough unlike other types of microscopes which have better resolution. This implies that the authors can easily miss the formation of new viral forms of CA derived from a partial uncoating at the pore. The authors should clarify this point.
We agree with the reviewer that fluorescence imaging does not provide sufficient spatial resolution for detecting structural changes in the conical HIV-1 core (not sure about the sensitivity, as fluorescence microscopy can detect single molecules). This information is now included in Discussion (lines 628-630).
3. In the experiments described in figure 1 and in figure 2 there is a condition missed. High dose of PF74. In fact, it has been described by the group of Yamashita that PF74 has a bimodal behavior dose dependent. The addition of this condition in both experiments will certainly strenghten the results.
We appreciate this suggestion, but have chosen to limit our experiments to low doses of PF74, in order to avoid dealing with the complexity of its effects on HIV-1 at higher doses. An additional effect of high-doses of PF74 on nuclear speckle localization of HIV-1 has been previously described [11]. A low PF74 dose was selected to block the nuclear import without affecting reverse transcription.
4. Line 530 the results indicating that HIV-1 nuclear import and clustering occur independently of reverse transcription confirmed results published by others that have not been mentioned, please introduce the ref here (Rensen et al., BioRxiv 2020).
We have now included the Rensen et al. paper in line 530, following the sentence: “Thus, consistent with recent findings, vDNA synthesis can continue beyond HIV-1 nuclear import and VRC clustering”.
5. The authors should better interpret and discuss the results showed in figure 6F. The % of co-localization of the vDNA with CDK9/pS175 is similar with or without RAL. However, episomal forms have a really low level of transcription. So how the authors can combine these results with their hypothesis that the detected EdU nuclear spots represent sites of active viral transcription?
We agree with the reviewer that in the absence of integration viral transcription is less efficient. We have recently reported [11] that only a few transcribed HIV-1 RNAs are detected in the nucleus after RAL treatment. Given the current data that the P-TEFb complex associates with nuclear HIV-1 DNA during RAL treatment, we hypothesize that, while transcription may occur from 2-LTR circles, it does so inefficiently. We have now corrected these sentences (lines 549-555) as follows:
“These results are in good agreement with our previous observation that a significantly lower number of vRNA puncta is associated with nuclear VRC clusters in the presence of RAL [11]. Taken together, these results suggest that, whereas P-TEFb may associate with nuclear HIV-1 clusters, vDNA integration into the host-genome is required for efficient viral transcription, in contrast to inefficient transcription from unintegrated vDNA templates in RAL-treated cells [19,20].”
Our observation that HIV-1 integration in macrophages also occurs in nuclear speckle associated genomic domains [11] further supports our hypothesis that integration may occur at the sites of HIV-1 nuclear clusters.
Discussion revision:
- Is it correct to write “in large part” in line 580? I could not find any quantitative data in the paper.
Pertinent quantitative data are displayed in Fig. 1C and Suppl. Fig. S2B, which show that a large fraction of integration-competent complexes enter the nucleus before the completion of vDNA synthesis, as determined by live-cell imaging and time of NVP addition, respectively. Also, Fig. 2 suggests that in MDMs HIV-1 escapes from PF74 inhibition (which included blocking nuclear import at 2 µM) much earlier than escape from NVP. We believe these data strongly suggest that a large fraction of HIV-1 complexes complete reverse transcription in the nucleus.
2. line 588 two references are missed (Selyutina et al., Cell Reports, 2020; Rensen et al., BioRxiv, 2020).
We thank the reviewer for pointing this out, these references are now cited in line 579.
3. Line 642: Correctly the authors discussed that rare viral nuclear events caracterized by intact cores have been detected inside the nucleus using sophisticated microscopy tools. Correctly the authors mentioned that these rare structures can be empty structure. However, the authors miss to discuss another recent paper (Blanco et al., JVI, 2020) that shows open CA structure chains containing the retrotranscribed genome translocating through the NPC.
We apologize for the omission and have now included the Blanco et al. paper in Discussion (lines 628-633).
“While fluorescence imaging lacks the resolution to study the conformational changes in the conical core, techniques, such as correlative fluorescence and electron microscopy (CLEM), have been successfully applied to gain structural insights into HIV-1 uncoating [6,16]. Collectively, these studies suggested that, while intact or nearly intact cores can be accommodated within the nuclear pore complex, their passage into the nucleoplasm results in dramatic restructuring of the viral core [6,16] and loss of the vRNP-associated electron density [16]. These studies provide proof of concept that partial CA assemblies might be retained after the loss of the integrity of CA lattice, however, the relevance of these CA-containing empty structures to infection remains unclear.”
4. These viral structures have been visualized using CLEM. In parallel the authors should discuss that several groups were able to visualize the viral CA associated with the viral DNA inside the nucleus (Peng et al., 2014). The biochemical presence of the viral CA protein was shown for the first time in the nucleus of macrophages and HeLa cells by Zhou and colleagues. The presence of CA in the nucleus was shown also by IF by several other groups (BejaranoDA, et al., 2019; Zurnic Bonisch I et al., 2020).
We apologize for the mishap in proper referencing. The papers are now cited in the beginning of the paragraph in a sentence (lines 623-625) that discusses detection of nuclear CA.
5. The discussion should be better organized as there are several repetitions that could be avoided to help the reader.
Again, we apologize for the inadvertent copy-paste errors during formatting of Discussion and thank the reviewer for catching this error. This has been corrected.
Literature Cited.
- Schroder, A.R.; Shinn, P.; Chen, H.; Berry, C.; Ecker, J.R.; Bushman, F. HIV-1 integration in the human genome favors active genes and local hotspots. Cell 2002, 110, 521-529.
- Campbell, E.M.; Hope, T.J. HIV-1 capsid: the multifaceted key player in HIV-1 infection. Nat Rev Microbiol 2015, 13, 471-483, doi:10.1038/nrmicro3503.
- Burdick, R.C.; Hu, W.S.; Pathak, V.K. Nuclear import of APOBEC3F-labeled HIV-1 preintegration complexes. Proc Natl Acad Sci U S A 2013, 110, E4780-4789, doi:10.1073/pnas.1315996110.
- Burdick, R.C.; Delviks-Frankenberry, K.A.; Chen, J.; Janaka, S.K.; Sastri, J.; Hu, W.S.; Pathak, V.K. Dynamics and regulation of nuclear import and nuclear movements of HIV-1 complexes. PLoS Pathog 2017, 13, e1006570, doi:10.1371/journal.ppat.1006570.
- Bejarano, D.A.; Peng, K.; Laketa, V.; Borner, K.; Jost, K.L.; Lucic, B.; Glass, B.; Lusic, M.; Muller, B.; Krausslich, H.G. HIV-1 nuclear import in macrophages is regulated by CPSF6-capsid interactions at the Nuclear Pore Complex. Elife 2019, 8, doi:10.7554/eLife.41800.
- Blanco-Rodriguez, G.; Gazi, A.; Monel, B.; Frabetti, S.; Scoca, V.; Mueller, F.; Schwartz, O.; Krijnse-Locker, J.; Charneau, P.; Di Nunzio, F. Remodeling of the Core Leads HIV-1 Preintegration Complex into the Nucleus of Human Lymphocytes. J Virol 2020, 94, doi:10.1128/JVI.00135-20.
- Francis, A.C.; Melikyan, G.B. Single HIV-1 Imaging Reveals Progression of Infection through CA-Dependent Steps of Docking at the Nuclear Pore, Uncoating, and Nuclear Transport. Cell Host Microbe 2018, 23, 536-548 e536, doi:10.1016/j.chom.2018.03.009.
- Mamede, J.I.; Cianci, G.C.; Anderson, M.R.; Hope, T.J. Early cytoplasmic uncoating is associated with infectivity of HIV-1. Proc Natl Acad Sci U S A 2017, 114, E7169-E7178, doi:10.1073/pnas.1706245114.
- Francis, A.C.; Marin, M.; Shi, J.; Aiken, C.; Melikyan, G.B. Time-Resolved Imaging of Single HIV-1 Uncoating In Vitro and in Living Cells. PLoS Pathog 2016, 12, e1005709, doi:10.1371/journal.ppat.1005709.
- Francis, A.C.; Di Primio, C.; Quercioli, V.; Valentini, P.; Boll, A.; Girelli, G.; Demichelis, F.; Arosio, D.; Cereseto, A. Second generation imaging of nuclear/cytoplasmic HIV-1 complexes. AIDS Res Hum Retroviruses 2014, 30, 717-726, doi:10.1089/AID.2013.0277.
- Francis, A.C.; Marin, M.; Singh, P.K.; Achuthan, V.; Prellberg, M.J.; Palermino-Rowland, K.; Lan, S.; Tedbury, P.R.; Sarafianos, S.G.; Engelman, A.N., et al. HIV-1 replication complexes accumulate in nuclear speckles and integrate into speckle-associated genomic domains. Nat Commun 2020, 11, 3505, doi:10.1038/s41467-020-17256-8.
- Borrenberghs, D.; Dirix, L.; De Wit, F.; Rocha, S.; Blokken, J.; De Houwer, S.; Gijsbers, R.; Christ, F.; Hofkens, J.; Hendrix, J., et al. Dynamic Oligomerization of Integrase Orchestrates HIV Nuclear Entry. Sci Rep 2016, 6, 36485, doi:10.1038/srep36485.
- Dharan, A.; Bachmann, N.; Talley, S.; Zwikelmaier, V.; Campbell, E.M. Nuclear pore blockade reveals that HIV-1 completes reverse transcription and uncoating in the nucleus. Nat Microbiol 2020, 10.1038/s41564-020-0735-8, doi:10.1038/s41564-020-0735-8.
- Hulme, A.E.; Kelley, Z.; Foley, D.; Hope, T.J. Complementary assays reveal a low level of CA associated with nuclear HIV-1 viral complexes. J Virol 2015, 10.1128/JVI.00476-15, doi:10.1128/JVI.00476-15.
- Peng, K.; Muranyi, W.; Glass, B.; Laketa, V.; Yant, S.R.; Tsai, L.; Cihlar, T.; Muller, B.; Krausslich, H.G. Quantitative microscopy of functional HIV post-entry complexes reveals association of replication with the viral capsid. Elife 2014, 3, e04114, doi:10.7554/eLife.04114.
- Zila, V.; Margiotta, E.; Turonova, B.; Müller, T.G.; Zimmerli, C.E.; Mattei, S.; Allegretti, M.; Börner, K.; Rada, J.; Müller, B., et al. Cone-shaped HIV-1 capsids are transported through intact nuclear pores. bioRxiv 2020, 10.1101/2020.07.30.193524, 2020.2007.2030.193524, doi:10.1101/2020.07.30.193524.
- Zila, V.; Muller, T.G.; Laketa, V.; Muller, B.; Krausslich, H.G. Analysis of CA Content and CPSF6 Dependence of Early HIV-1 Replication Complexes in SupT1-R5 Cells. mBio 2019, 10, doi:10.1128/mBio.02501-19.
- Zurnic Bonisch, I.; Dirix, L.; Lemmens, V.; Borrenberghs, D.; De Wit, F.; Vernaillen, F.; Rocha, S.; Christ, F.; Hendrix, J.; Hofkens, J., et al. Capsid-Labelled HIV To Investigate the Role of Capsid during Nuclear Import and Integration. J Virol 2020, 94, doi:10.1128/JVI.01024-19.
- Sloan, R.D.; Wainberg, M.A. The role of unintegrated DNA in HIV infection. Retrovirology 2011, 8, 52, doi:10.1186/1742-4690-8-52.
- Gillim-Ross, L.; Cara, A.; Klotman, M.E. HIV-1 extrachromosomal 2-LTR circular DNA is long-lived in human macrophages. Viral Immunol 2005, 18, 190-196, doi:10.1089/vim.2005.18.190.
Reviewer 2 Report
The manuscript entitled "HIV-1 uncoating and nuclear import precede the completion of reverse transcription in cell lines and in primary macrophages" authored by Ashwanth C. Francis and colleagues is well written and suitable for publication.
Minor points
Introduction, results and figures are redundant.
Authors performed the experiments using plasmids and pseudoviruses. They should use HIV strains to infect cells and electron microscopy to confirm their data.
Author Response
Reviewer-2:
The manuscript entitled "HIV-1 uncoating and nuclear import precede the completion of reverse transcription in cell lines and in primary macrophages" authored by Ashwanth C. Francis and colleagues is well written and suitable for publication.
Minor points
Introduction, results and figures are redundant.
We thank the reviewer for this comment and have edited and reformatted the manuscript accordingly.
Authors performed the experiments using plasmids and pseudoviruses. They should use HIV strains to infect cells and electron microscopy to confirm their data.
Pseudoviruses are a widely accepted and validated system to study HIV-1 and other virus entry, including uncoating and nuclear import of HIV-1 complexes. Besides biosafety concerns when using live cell microscopy, infectious virus do not offer any tangible advantages for examining the early steps of viral infection. As to the electron microscopy, while this is a powerful technique, it is not suitable for live cell imaging and functional studies of virus entry. For this reason, we did not rely on EM to study HIV-1 uncoating and nuclear import. However, we are currently working on implementing a correlative light-electron microscopy (CLEM) technique which would allow structural studies of HIV-1 core uncoating in the future.
Reviewer 3 Report
This is a well-performed study in relevant cell types. I have only one major concern that needs to be addressed either experimentally, or at least through acknowledgement in the text:Virtually all the imaging data starts at times that would be considered quite late in the initial stages of infection, focusing ONLY on capturing late uncoating events at the nuclear membrane. There are two related major issues with this approach. First, how do the authors know that the particles that have reached the nuclear membrane that they are imaging did not undergo cytoplasmic uncoating before they arrived there? Second, how do the authors know that the viral cores that are at the nuclear membrane are fully intact as opposed to already partially uncoated?
These are fundamental issues that the authors need to address or at least acknowledge in the discussion to give a more balanced view on the limitations with their imaging approaches and their focus on later events that do not account for what might have happened prior to imaging.
Author Response
This is a well-performed study in relevant cell types. I have only one major concern that needs to be addressed either experimentally, or at least through acknowledgement in the text:
Virtually all the imaging data starts at times that would be considered quite late in the initial stages of infection, focusing ONLY on capturing late uncoating events at the nuclear membrane.
We thank the reviewer for this comment, but would like to point out that we did image early post-fusion events in MDMs and other cells. Fig. 3A, B examines the extent of CDR retention by cores at 80 min post-infection. This is a rather early time point for MDMs in which HIV-1 infection proceeds for several days. In addition, Suppl. Fig. S4 shows images and tracks of early uncoating events that are manifested by a loss of CDR signal down to a background level and a subsequent loss of the INsfGFP signal, apparently due to proteasomal degradation [7].
There are two related major issues with this approach. First, how do the authors know that the particles that have reached the nuclear membrane that they are imaging did not undergo cytoplasmic uncoating before they arrived there? Second, how do the authors know that the viral cores that are at the nuclear membrane are fully intact as opposed to already partially uncoated?
These are fundamental issues that the authors need to address or at least acknowledge in the discussion to give a more balanced view on the limitations with their imaging approaches and their focus on later events that do not account for what might have happened prior to imaging.
We agree with this point and would like to stress that we have never claimed that HIV-1 cores docked at the nuclear envelope are intact. On the contrary, both in Introduction (lines 74-85) and in Discussion (lines 611-617), we state the existence of two types of uncoating events – drastic and complete loss of CDR early after viral fusion, which leads to VCR degradation in the cytoplasm [7,9], and gradual uncoating of the remainder of HIV-1 cores that later dock and uncoat at the nuclear pore, which in cell lines leads to productive infection [7]. Furthermore, in Discussion, we explicitly state (lines 620-621) and provide appropriate references that partial uncoating might be required for core trafficking through the cytoplasm and interactions with the NPC components. What we conclude is that the terminal step manifested in a marked loss of CDR/CA occurs at the nuclear membrane. In order to avoid confusion and per the reviewer’s suggestion, we added a sentence (lines 617-620) acknowledging the limitations of live cell single virus imaging techniques.